# Sesquiterpene Lactones with Anti-Inflammatory Activity from the Halophyte *Sonchus brachyotus* DC

**DOI:** 10.3390/molecules28041518

**Published:** 2023-02-04

**Authors:** Young-Kyung Lee, Hangy Lee, Yun Na Kim, Jun Kang, Eun Ju Jeong, Jung-Rae Rho

**Affiliations:** 1Department of Oceanography, Kunsan National University, Jeonbuk 54150, Republic of Korea; 2Department of Plant & Biomaterials Science, Gyeongsang National University, Jinju 52725, Republic of Korea; 3Department of Marine Biotechnology, Kunsan National University, Jeonbuk 54150, Republic of Korea

**Keywords:** *Sonchus brachyotus*, eudesmanolide, sesquiterpene lactone, anti-inflammation

## Abstract

There were five sesquiterpene lactones, belonging to the eudesmanolide class, isolated from the halophyte *Sonchus brachyotus* DC. The structures of the compounds were determined using spectroscopic methods, including 1D and 2D NMR spectra, MS data, and optical rotation values. Compounds **4** and **5** were characterized by the position of *p*-hydroxyphenylacetyl group in the sugar moiety. In the evaluation of anti-inflammatory effects on LPS-activated RAW264.7 macrophages, compound **1**, 5α,6βH-eudesma-3,11(13)-dien-12,6α-olide, potently suppressed the expression of iNOS and COS-2, as well as the production of TNF-α, IL-6, and IL-10. Treatment of **1** regulates the Nrf2/HO-1 pathway.

## 1. Introduction

*Sonchus* species are known as annual, biennial, and perennial herbaceous plants [1]. Their aerial parts are prolific sources of proteins, vitamins, carbohydrates, minerals, and essential amino acids [2]. Furthermore, biological active chemical constituents [3,4,5,6,7], including steroid, sesquiterpene lactones, flavonoids, and glycosides, are isolated from *Sonchus* species and exhibit pharmaceutical properties, such as anticancer, antioxidant, anti-inflammatory, and antibacterial activities [8,9,10,11]. In recent years, *Sonchus* species have attracted attention as promising sources of functional foods or pharmaceuticals [12]. 

In our continuing search for bioactive metabolites from halophyte plants, the methanol extract of *S. brachyotus* from the western coast of Korea showed anti-inflammatory effects in RAW264.7 cells. Bioassay-guided separation and chemical investigation of the extract using successive column chromatography led to the isolation of three new and two known eudesmanolide compounds. Here, we report the structure determination of the new compounds by spectroscopic methods and biological evaluation. 

## 2. Results and Discussion

### 2.1. Isolation of Sesquiterpenes (***1***–***5***) from Sonchus Brachyotus

The methanolic extract from the whole plant of *S. brachyotus* was partitioned between butanol and H_2_O, and the butanol layer was separated by reversed silica column chromatography and HPLC to yield two known (**1** and **3**) and three new (**2, 4,** and **5**) eudesmanolides (Figure 1).

Compounds **1** and **3** were identified as (1*R*, 5*R*, 6*R*, 7*S*, 10*S*)-5α,6βH-eudesma-3, 11(13)-dien-12,6α-olide and 1β-*O*-β-*D*-glucopyranosyl-(1*R*, 5*R*, 6*R*, 7*S*, 10*S*)-5α,6βH-eudesma-3, 11(13)-dien-12,6α-olide, respectively, by interpretation of 1D and 2D NMR spectra and comparison with optical rotation values reported in the literature [13,14]. 

Compound **2** was obtained as a colorless gum, and its molecular formula was suggested to be C_21_H_30_O_9_ from the HRESI mass spectrum and the ^13^C NMR spectrum. The ^1^H and ^13^C NMR spectra displayed signals similar to **1**. On comparing the ^1^H NMR spectrum and molecular formula with those of **1**, it was found that **2** includes a sugar moiety. The sugar was linked to C-1, based on the HMBC correlation between the anomeric proton (δ_H_ 4.31) and C-1 (δ_C_ 81.2). Based on the NOESY spectrum and proton coupling constants in the ^1^H NMR spectrum, the configuration of the aglycon part of **2** was identical to those of **1** and **3**. The sugar moiety was assigned as the β-*D*-form based on the coupling constant (*J*_1-H′,2-H′_ = 7.8 Hz) of the anomeric proton and the measured optical rotation value ([α]^20^_D_~+50°, *c* 0.1, H_2_O) after acidic hydrolysis of **2**. Thus, **2** was assigned as 1β-*O*-β-*D*-glucopyranosyl-5α,6βH-eudesma-3,11(13)-dien-12,6α-olide. 

Compound **4** was found to have the molecular formula C_29_H_36_O_11_, based on the fragment of [M+NH_4_]^+^ from its positive HRESI mass spectrum and its ^13^C NMR spectrum, corresponding to 12 degrees of unsaturation. The IR and UV spectra displayed absorption peaks at 3384, 1746, 1513, 1265, and 1081 cm^−1^ and 203 and 276 nm, respectively, indicating the presence of a phenyl group. The ^13^C NMR signals were similar to those of **3** except for the signals at 123.1 and 137.4 ppm, while the ^1^H NMR signals for the two compounds differed in the range of 0.5~4.5 ppm. From the 1D and 2D NMR interpretations, the difference was attributed to the position of the *p*-hydroxyphenylacetate unit in **4**. The unit was connected to C-2′, which was revealed by the HMBC correlation between H-2′ (δ_H_ 4.66) and C-8‴ (δ_C_ 172.8). This was supported by a deshielded chemical shift of H-2′, resulting from an ester bond with the *p*-hydroxyphenylacetyl unit. The coupling constants, from H-1′ to H_2_-6′, was recognized as β-glucopyranose. In a similar manner, the carbohydrate was determined to be *D*-form by optical rotation value, [α]^20^_D_ = 51°, after acidic hydrolysis of **4**. The configuration of the aglycon skeleton of **4** was identical to that of **1**, as established by the ROESY spectrum. In addition, the ROE correlations between H-2‴/-6‴ and H-14, H-3‴/-5‴ and H_3_-14, H-2‴/-6‴ and H-2β, H-2α and H-1′, as well as H-2β and H-1′,, suggest the dominant conformation of **4** (Figure 2). The ring current of the *p*-hydroxyphenylacetate group bonded to C-2′ caused H-2β (δ_H_ 1.44) and H_3_-14 (δ_H_ 0.67) to shift to the upfield region. Accordingly, compound **4** was assigned as 1β-*O*-β-*D*-glucopyranosyl-(2′-*O*-*p*-hydroxyphenylacetyl)-(1*R*, 5*R*, 6*R*, 7*S*, 10*S*)-5α,6βH-eudesma-3, 11(13)-dien-12,6α-olide. 

Compound **5** was identified as C_37_H_42_O_13_ based on the fragment of [M+NH_4_]^+^ at 712.2951 from its positive HRESI mass spectrum and ^13^C NMR spectrum. The IR and UV spectra were identical to those of **4**. Comparing them with the molecular formula and the ^13^C NMR spectrum, **5** was deduced to possess one more *p*-hydroxyphenylacetate unit than **4**. The two *p*-hydroxyphenylacetate units were connected to C-15 and C-2′, as revealed by the HMBC correlations of H-2′(δ_H_ 4.40) and C-8‴(δ_C_ 172.7) and of H_2_-15 (δ_H_ 4.39 and 4.61) and C-8″(δ_C_ 173.2). Due to of the ring current from the two *p*-hydroxyphenylacetate units, most of the proton chemical shifts of **5** shifted upfield. In particular, the proton chemical shifts of H-2β (δ_H_ 1.34) and H_3_-14 (δ_H_ 0.45) showed major differences compared with those (δ_H_ 2.10 and 0.93) of **2**. Thus, compound **5** was established as 1β-*O*-β-*D*-glucopyranosyl-(2′-*O*-*p*-hydroxyphenylacetyl)-15-*O*-(*p*-hydroxyphenylacetyl)-5α,6βH-eudesma-3,11(13)-dien-12,6α-olide.

### 2.2. Cytotoxicities and Anti-Inflammatory Activities of ***1***–***5*** in Lipopolysaccharide-Activated RAW264.7

Based on the finding that the methanolic extract of *S. brachyotus* effectively reduced the production of nitric oxide (NO) in lipopolysaccharide (LPS)-activated RAW264.7, the anti-inflammatory activities of the isolated compounds **1**–**5** were evaluated in the same in vitro system. Prior to screening the activities of **1**–**5**, their cytotoxicity was measured using the MTT assay (Figure 3). Compounds **1**–**5** showed no cytotoxicity at the concentrations tested (10 and 20 µM), while the treatment of RAW264.7 cells with **1** resulted in the increased cell viability at a concentration of 20 µM (Figure 3A). To evaluate the inhibitory activities of **1**–**5** on NO production, RAW264.7 cells were treated with each compound (10 μM) for 1 h and, then, treated with LPS (100 ng/mL). After 24 h of incubation, the level of NO in the culture medium was detected using Griess reagent. It was observed that the content of NO induced by LPS decreased in cells treated with **1**, **2**, or **4**. The treatment of cells with **1** led to a significant decrease in NO production (Figure 3B). 

Increased levels of pro-inflammatory cytokines, such as TNF-α, IL-1β, IL-6, and IL-10, are observed during inflammation and immune responses. The effects of compounds **1**–**5** on the production of these inflammatory mediators were measured using ELISA (Figure 4). The levels of IL-10, IL-6, and TNF-α induced by LPS were significantly decreased by treatment with compounds, except for **2**. Similar to the NO content, pretreatment of cells with **1** led to a significant decrease in LPS-induced IL-10, IL-6, and TNF-α production.

### 2.3. Effects of ***1*** from S. Brachyotus on the Expression of Pro-Inflammatory Proteins and MAPK Phosphorylation in Lipopolysaccharide-Activated RAW264.7

The expressions of inducible nitric oxide synthase (iNOS) or cyclooxygenase (COX-2) are not found in most resting cells. The exposure of cells to endogenous and exogenous stimulants, such as LPS, interleukins, TNF-α, or interferon, is known to trigger the expression of these pro-inflammatory proteins. Overexpression of iNOS and COX-2 induces the production of NO and inflammatory cytokines in activated macrophages [15,16]. Based on the activities of compounds **1**–**5** on NO and cytokine production, the anti-inflammatory actions of **1**, the most bioactive compound, were further investigated. As shown in Figure 5 and Figure 6, LPS-induced up-regulation of iNOS and COX-2 by LPS were significantly reduced by **1** (5, 10, and 20 μM). In addition, the increased phosphorylation of members of the mitogen-activated protein kinase (MAPK) family, including p38, ERK1/2, and JNK induced by LPS, was inhibited in cells treated with **1**. The phosphorylation of ERK1/2 and p-38 was attenuated at a concentration of 20 µM of **1**, whereas the phosphorylation of JNK was attenuated by **1** in a concentration-dependent manner. Though nonsteroidal anti-inflammatory drugs NSAIDs have long been used to treat pain and inflammation, various side effects have been reported, including bronchospasm, ulcers, ear pain, and water retention in the body. Regarding side effects caused by NSAIDs, it has been recently theorized that COX inhibitors, such as fenamates, can interact with phospholipid membranes to alter the structure, dynamics, and fluidity of membranes and, consequently, affect non-cognate receptors, resulting in protein dysfunction [17].

### 2.4. Effects of 1 from S. brachyotus on HO-1 Expression Mediated by Nuclear Translocation Nrf2 in Lipopolysaccharide-Activated RAW264.7

Heme oxygenase (HO)-1 is an inducible isoform of the heme-degrading enzyme HO. HO-1 plays a crucial role in the inflammation process in which HO-1 suppresses the expression of pro-inflammatory mediators in activated macrophages [18]. Nuclear factor E2-related factor 2 (Nrf2) is a transcription factor that protects against oxidative stress by binding to antioxidant response elements (ARE) located in the promoters of genes encoding antioxidant enzymes, including HO-1 [19]. It is well known that HO-1/Nrf2 signaling is induced by various stimuli via the activation of the MAPK signaling pathway. To determine whether **1** regulates Nrf-2/HO-1 signaling, the expression level of HO-1 and the nuclear and cytoplasmic levels of Nrf2 in cells treated with **1** were measured. As shown in Figure 7 and Figure 8, HO-1 and Nrf2 protein levels were increased by treatment with **1**. The nuclear level of Nrf2 increased by **1** in a concentration-dependent manner, and the cytoplasmic level of Nrf2 showed a concomitant decrease. These results suggest that treatment of RAW264.7 cells with **1** promoted the nuclear translocation of Nrf2 and increased the expression of HO-1.

## 3. Materials and Methods

### 3.1. Instrumentation

The optical rotation values were measured using a JASCO P-1010 polarimeter (Jasco, Easton, MD, USA). IR spectra were recorded on a JASCO FT/IR 4100 spectrometer (Jasco, Easton, MD, USA), and ultraviolet (UV) spectra were recorded using a Varian Cary 50 UV-visible spectrophotometer (Agilent, Santa Clara, CA, USA). High-resolution (HR)-electrospray ionization (ESI) mass spectra were measured using a SCIEX X500R mass spectrometer (Sciex, Framingham, MA, USA). Nuclear magnetic resonance (NMR) spectra were recorded on a Varian VNMRS 500 NMR spectrometer (Varian, Palo Alto, CA, USA), operating at 500 MHz (^1^H) and 125 MHz (^13^C), with chemical shifts of the proton and carbon spectra measured in methanol-*d*_4_ solution, were reported in reference to residual solvent peaks at 3.30 ppm and 49.0 ppm, respectively. Semi-preparative liquid chromatography (SemiPrep-LC) was performed using a Waters 515 pump (Agilent, Santa Clara, CA, USA) equipped with an RI detector. Column chromatography was performed using an RP-18 silica gel 60 (Merck, Darmstadt, Germany) and Sephadex LH-20 (Pharmacia, Uppsala, Sweden).

### 3.2. Material 

*S. brachyotus* DC was collected from salterns in Taean and Gochang, South Korea in 2019. Plant identification was authenticated by Prof. Min Hye Yang (Pusan National University, Korea). A voucher specimen was deposited at the Laboratory of Marin Natural Product Chemistry, Kunsan National University, Republic of Korea.

### 3.3. Extraction and Isolation 

The dried sample was extracted twice with methanol for two days and partitioned between butanol and water. The organic layer was subjected to reverse-phase silica flash chromatography to give seven fractions, which were eluted with 10% increments of methanol from 50% methanol and 100% acetone: MR1–MR7. Fractions MR2 and MR3 were selected based on their activity and ^1^H NMR spectra. First, MR2 was chromatographed on a Sephadex LH20 column eluted with 100% methanol to yield four subfractions: M1–M4. Subfraction M2 was separated by reverse-phase silica HPLC to yield four compounds: **2** (5.0 mg) at a retention time of 14 min, **1** (2.4 mg) at 22 min, **3** (3.0 mg) at 25 min, and **4** (8.6 mg) at 26 min. HPLC was performed as follows: isocratic elution with 30% ACN and 70% H_2_O with 0.1% formic acid, YMC ODS A column (250 × 10 mm), a flowrate of 1.2 mL/min, and a refractive index detector. Compound **2** was further purified by HPLC as follows: isocratic elution with 40% MeOH and 60% H_2_O with 0.1% formic acid, Phenomenex polar column (250 × 10 mm), a flowrate of 2.0 mL/min, and UV detector (210 nm). Second, compound **5** (1.8 mg) was isolated at a retention time of 48 min by chromatography from fraction MR3. Similarly, HPLC was performed using a Phenomenex polar RP column (250 × 10 mm) and 2.0 mL/min flow, with the same eluting solvent and detector as the previous one. 

The 1β-*O*-β-*D*-glucopyranosyl-5α,6βH-eudesma-3,11(13)-dien-12,6α-olide (**2**) is amorphous oil. [α]^20^_D_ + 19 (*c* 0.1, MeOH). UV (MeOH) λ_max_ (log ε): 203 (4.3) nm. IR (film) ν_max_: 3348, 2921, 1748, 1594, and 1076 cm^−1^. Additionally, ^1^H (500 MHz) and ^13^C (125 MHz) NMR data are listed in Appendix A. HRESIMS (positive-ion mode) m/z: 444.2226 [M+NH_4_]^+^ (calculated for C_21_H_34_NO_9_^+^, 444.2228, Δ = 0.5 ppm).

The 1β-*O*-β-*D*-glucopyranosyl-(2′-*O*-*p*-hydroxyphenylacetyl)-5α,6βH-eudesma-3,11(13)-dien-12,6α-olide (**4**) is amorphous oil. [α]^20^_D_ +50 (*c* 0.1, MeOH). UV (MeOH) λ_max_ (log ε): 203 (4.3) and 276 (3.4) nm. IR (film) ν_max_: 3384, 2928, 1746, and 1265 cm^−1^. Additionally, ^1^H (500 MHz) and ^13^C (125 MHz) NMR data are listed in Table 1. HRESIMS (positive-ion mode) m/z: 578.2609 [M+NH_4_]^+^ (calculated for C_29_H_40_NO_11_^+^, 578.2596, Δ = 2.3 ppm).

The 1β-*O*-β-*D*-glucopyranosyl-(2′-*O*-*p*-hydroxyphenylacetyl)-15-*O*-(p-hydroxyphenylacetyl)-5α,6βH-eudesma-3,11(13)-dien-12,6α-olide (**5**) is an amorphous oil. [α]^20^_D_ +66 (*c* 0.15, MeOH). UV (MeOH) λ_max_ (log ε): 201 (4.1) and 278 (3.2) nm. IR (film) ν_max_: 3356, 2924, 1739, and 1265 cm^−1^. Additionally, ^1^H (500 MHz) and ^13^C (125 MHz) NMR data are listed in Table 1. HRESIMS (positive-ion mode) m/z: 712.2951 [M+NH_4_]^+^ (calculated for C_37_H_46_NO_13_^+^, 712.2964, Δ = 1.8 ppm).

### 3.4. NMR Experiments 

The 1D and 2D NMR spectra were obtained on a Varian VNMRS system working at 500MHz for proton and 125 MHz for carbon. The ^1^H and ^13^C NMR chemical shifts refer to CD_3_OD at 3.30 and 49.0 ppm, respectively. For all experiments, the temperature was stabilized at 297 K. The parameters used for the 2D NMR spectra are as follows: the gradient COSY spectra were collected with a relaxation delay of 1 *s* and a spectral width of 4000 Hz in a 512 (*t*1) × 1024 (*t*2) matrix, applying a pulse gradient of 1 *ms* duration with a strength of 10 G/m and processed with a sinebell function. The gradient HSQC spectra were measured with the phase-sensitive mode, the parameters of ^1^*J*_CH_ = 140 Hz, and a relaxation delay of 1 *s*, and they were processed with Gaussian function in a 256 (*t*1) × 1024 (*t*2) matrix by a linear prediction method for a higher resolution. The gradient HMBC spectra were performed with the absolute mode and the parameters of ^1^*J*_CH_ = 140 Hz, ^n^*J*_CH_ = 8 Hz, and a relaxation delay of 1 *s* under the pulse gradient of 1 *ms*, duration with a strength of 10 G/m, and processed with a sinebell function in a 512 (*t*1) × 1024 (*t*2) matrix. The ROESY spectra were measured with a spin-locking time of 350 *ms* and a scan number of 32.

### 3.5. Acdic Hydrolysis 

Compound **2** (1.5 mg) was treated with 35% HCl at room temperature for 1 h at 30 °C and neutralized with NaOH. After filtration and evaporation to dryness, the dried product was dissolved in a mixed solvent (CHCl_3_:MeOH:H_2_O = 15:3:1, 1 mL) and evaporated to dryness. Next, H_2_O was added to yield the sugar moiety (ca. 0.2 mg). The acid hydrolysis of **4** was performed in the same manner. 

### 3.6. Cell Cultures

RAW264.7 macrophage cells were obtained from the Korea Cell Line Bank (Seoul, Republic of Korea). The cells were maintained in DMEM containing 20 mM HEPES, 2 mM L-glutamine, 10% FBS with penicillin (100 IU/mL), and streptomycin (10 mg/mL) at 37 °C in a humidified atmosphere of 95% air-5% CO_2_.

### 3.7. Determination of NO Content

RAW264.7 cells were seeded in 48 well plates (1 × 10^5^ cells/well) and incubated at 37 °C for 24 h. Then, the cell culture was washed, and the medium was replaced with Griess medium to remove any trace of phenol red and treated with the test sample for 1 h before exposure to 0.1 mg/mL. After 24 h incubation, nitrite in culture medium was measured to assess NO production in RAW264.7 cells using a Griess reagent. The absorbance at 550 nm was measured using a microplate reader, and the concentration was determined using a nitrite standard curve.

### 3.8. Estimation of Cell Viability

Cell viability was determined using a colorimetric 3-(4,5-dimethylthiazol-2-yl)-2,5-diphenyl tetrazoliumbromide (MTT) assay based on the reduction in MTT (Sigma, St. Louis, MO, USA) to formazan by cellular dehydrogenase. After 100 μL of sample aliquot was collected for the Griess assay, MTT (0.2mg/mL) was directly added to cultures, followed by incubation at 37 °C for 2 h. The supernatant was aspirated and 100 µL of DMSO was added to dissolve the formazan. After the insoluble crystals were completely dissolved, the absorbance was measured at 540 nm using a microplate reader. Data are expressed as the percentage of cell viability relative to the control cultures.

### 3.9. Enzyme-Linked Immuno-Sorbent Assay (ELISA)

Determination of TNF-α, IL-6 and IL-10 production RAW264.7 cells were plated overnight in 96 well plates at a density of 2 × 10^4^ cells/well. The cells were treated with the samples to be tested for 1 h before exposure to 10 ng/mL (TNF-α, IL-6) or 100 ng/mL (IL-10) LPS. After incubation for 3 h (TNF-α), 6 h (IL-6), and 18 h (IL-10), supernatants were collected and used for cytokine measurement. Finally, the cellular levels of TNF-α, IL-6, and IL-10 were detected in the medium using the corresponding ELISA kits (BD Biosciences, Pharmingen, San Diego, CA, USA) according to the manufacturer’s instructions. The optical density was measured at 450 nm, and the amount of cytokines or chemokines was calculated from a standard curve prepared with the recombinant protein. The experiments were independently repeated at least thrice.

### 3.10. Preparation of Total Cell Extract

RAW264.7 cells were plated overnight in six-well plates at a density of 1 × 10^6^ cells/well. The medium was replaced with fresh medium, and the test sample was treated for 1 h before exposure to 0.1 µL/mL. After 24 h of incubation, cells were washed twice with phosphate buffered saline (PBS), and cell lysates were extracted with a lysis buffer (M-PER™ Mammalian Protein Extraction Reagent 78501, Thermos Scientific, Waltham, MA, USA) containing a protease inhibitor cocktail (Thermo Scientific, USA, Prod # 78425). The protein extracts were centrifuged at 13000 rpm for 20 min at RT. The supernatant was transferred to a pre-chilled tube.

### 3.11. Isolation of Nuclear and Cytoplasmic Extract

Nuclear extraction was performed using an NE-PER Nuclear Cytoplasmic Extraction Reagent kit (Pierce, Rockford, IL, USA) according to the manufacturer’s instructions. Briefly, treated cells were harvested and centrifuged at 1000 rpm for 3 min. The cell pellet was resuspended in 100 µL of cytoplasmic extraction reagent I by vortexing. The suspension was incubated on ice for 10 min, followed by the addition of 5.5 µL of a second cytoplasmic extraction reagent II, vortexed for 5 s, incubated on ice for 1 min, and centrifuged for 5 min at 12,000 rpm. The supernatant (cytoplasmic extract) was transferred to a pre-chilled tube. The insoluble pellet fraction, which contained crude nuclei, was resuspended in 50 µL of nuclear extraction reagent by vortexing for 15 s, incubating on ice for 10 min, and centrifuging for 10 min at 12,000 rpm. The resulting supernatant, which constituted the nuclear extract, was used for subsequent experiments.

### 3.12. Western Blotting

Total cell lysates and nuclear and cytoplasmic proteins were isolated as described above. The protein content of cell lysates was quantified using the Bradford assay. There were 30 micrograms of harvested proteins separated using 8–10% SDS-PAGE at 100V and transferred to polyvinylidene fluoride (PVDF) membranes. The membrane was blocked with 5% skimmed milk for 1 h at 25 °C. The membranes were incubated with 1:1000 diluted primary antibodies (Cell Signaling Technology, Inc., Danvers, MA, USA, Santa Cruz Biotechnology) at 4 °C overnight. After washing three times with TBST, immunoreactive bands were visualized using immunopure peroxidase conjugated mouse anti-rabbit IgG and goat anti-mouse IgG (1:10,000 dilution; Santa Cruz Biotechnology). Membranes were incubated with secondary antibodies for 1 h at 25 °C. The blots were washed three times with TBST buffer. Protein bands were visualized using ECL solution (Bio-Rad clarity Max western ECL substrate) and calibrated using the ChemiDoc Imaging System (Fusion FX5, Vilber Lourmat, Collégien, France). The density value of the protein bands was normalized to that of Lamin B (nuclear), β-actin, and α-tubulin (total protein or cytosol).

## 4. Conclusions

There were three new sesquiterpene lactones, together with two ones, isolated from the halophyte *S. brachyotus*, which lives in a saltern area. All compounds were identified as derivatives of the eudesmanolide skeleton. The connection of a glucose unit to the C-1 position of **1** led to the structure of **2**, named as 1β-glucospyranosyl-5α,6βH-eudesma-3,11(13)-dien-12,6α-olide, which has not been reported. In the three compounds, one or two *p*-hydroxyphenylaceylate groups were added to compound **2**. Compounds **4** and **5** were characterized by the linkage of a *p*-hydroxyphenylaceylate group on C-2′ of **2**, while similar compounds isolated from *Sonchus* species were commonly positioned on C-6′ [14,20]. 

The isolated sesquiterpenes were tested for their anti-inflammatory activity in LPS-activated RAW264.7 macrophages. Compound **1** showed the most potent activity. Treatment of RAW264.7 with **1** down-regulated the expression of pro-inflammatory proteins, iNOS, and COX-2, as well as the production of cytokines, TNF-α, IL-6, and IL-10. Phosphorylation of the MAPK family, ERK1/2, p-38, and JNK was decreased, and Nrf2/HO-1 signaling was regulated. These results suggest that the native halophyte *S. brachyotus* containing bioactive sesquiterpenes is a useful therapeutic agent for the treatment of inflammation-related diseases.

## Figures and Tables

**Figure 1 molecules-28-01518-f001:**
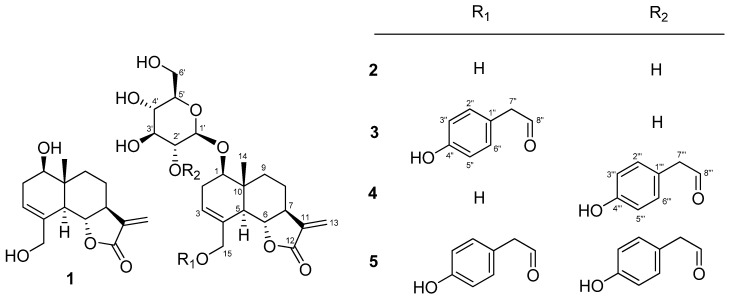
Structures of compounds **1–5**.

**Figure 2 molecules-28-01518-f002:**
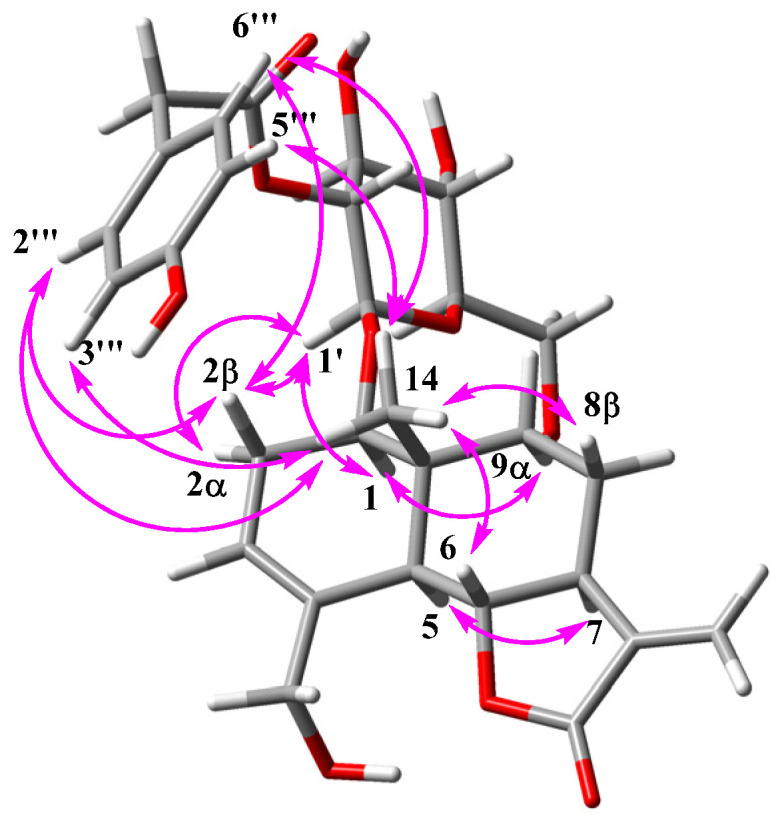
Key ROE correlations of **4**.

**Figure 3 molecules-28-01518-f003:**
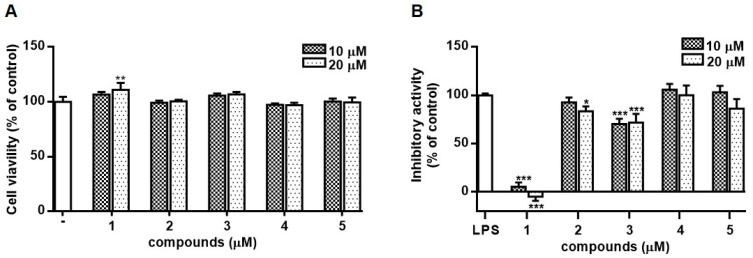
Effects of compounds **1**–**5** isolated from *S. brachyotus* on cell viability (**A**) and NO production (**B**) in RAW264.7 cells. RAW264.7 cells were incubated with each compound at the concentrations of 10 or 20 µM for 24 h. Cell viability was measured using the MTT assay.  The concentration of nitrite in the culture medium was measured by Griess reaction, and sodium nitrite was used as a standard. Results are presented as the mean ± standard deviation from three independent experiments. The results of cell viability (**A**) differ significantly from non-treated control cells, and the results of NO production (**B**) differ significantly from LPS-only treated cells; * *p* < 0.05, ** *p* < 0.01 and *** *p* < 0.001.

**Figure 4 molecules-28-01518-f004:**
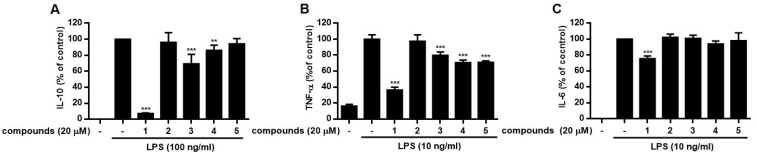
Effects of compounds **1**−**5** isolated from *S. brachyotus* on IL-10 (**A**), TNF-α (**B**), and IL-6 (**C**) production in LPS-stimulated RAW264.7 cells. The cells were treated with each compound (20 μM) for 1 h before exposure to 10 ng/mL (TNF-α, IL-6) or 100 ng/mL (IL-10) of LPS. After incubation for 3 h (TNF-α), 6 h (IL-6), and 18 h (IL-10), the levels of IL-10, TNF-α, and IL-6 in the culture medium were subsequently measured using commercially available ELISA kits. Results are presented as the mean ± standard deviation from three independent experiments. The results differ significantly from LPS-only treated; ** *p* < 0.01, *** *p* < 0.001.

**Figure 5 molecules-28-01518-f005:**
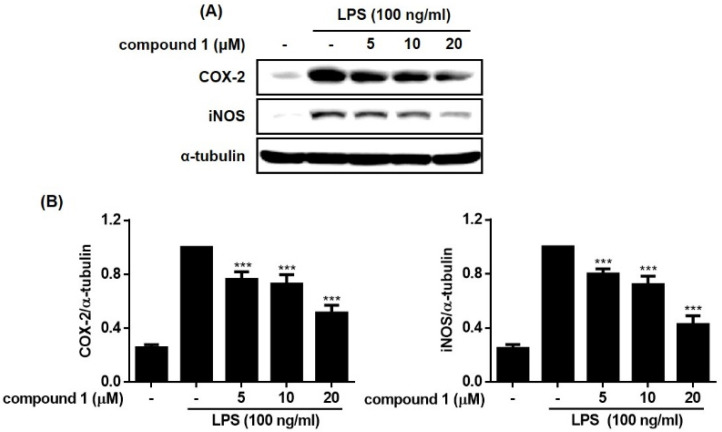
Compound **1** down-regulated the expression of inducible nitric oxide synthase (iNOS) and cyclooxygenase 2 (COX-2) in RAW264.7 cells stimulated with LPS. The RAW264.7 cells were pretreated with the compound (5, 10, and 20 μM) for 1h before exposure to LPS for 24 h. The representative blots (**A**) and the calculated intensities (**B**) after normalization to α-tubulin were presented. Results are presented as the mean ± standard deviation from three independent experiments. The results differ significantly from LPS-only treated; *** *p* < 0.001.

**Figure 6 molecules-28-01518-f006:**
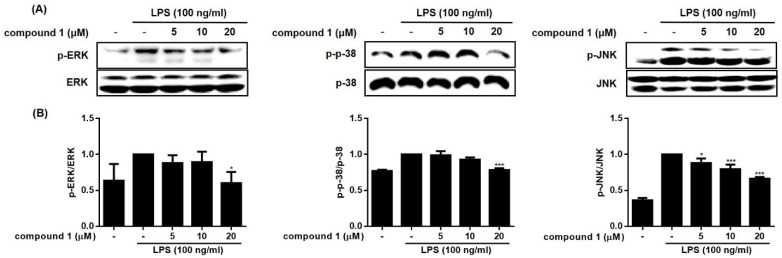
Compound **1** attenuated the phosphorylation of mitogen-activated protein kinase (MAPK) family in RAW264.7cells. The RAW264.7 cells were pretreated with the compound (5, 10, and 20 μM) for 1 h before exposure to LPS for 24 h. The phosphorylated MAPKs (JNK, ERK, and p38) were analyzed using Western blotting. The representative blots (**A**) and the calculated intensities after normalization to α-tubulin (**B**) were presented. Results are presented as the mean ± standard deviation from three independent experiments. The results differ significantly from LPS-only treated; * *p* < 0.05, *** *p* < 0.001.

**Figure 7 molecules-28-01518-f007:**
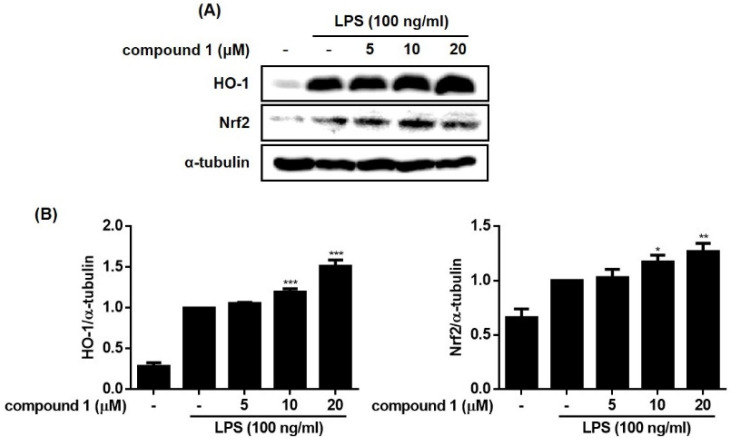
Effects of compound **1** on Nrf2/HO-1 pathway in LPS-induced RAW264.7 cells. The RAW264.7 cells were pretreated with the compound (5, 10, and 20 μM) for 1h before exposure to LPS for 24 h. The representative blots (**A**) and the calculated intensities after normalization to α-tubulin (**B**) were presented. Results are presented as the mean ± standard deviation from three independent experiments. The results differ significantly from LPS-only treated; * *p* < 0.05, ** *p* < 0.01, and *** *p* < 0.001.

**Figure 8 molecules-28-01518-f008:**
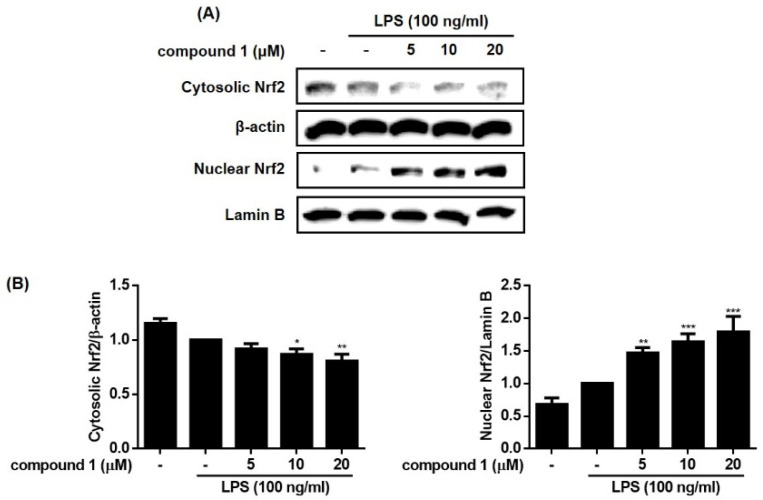
Effects of compound **1** on Nrf2 protein expression in RAW264.7cells. The RAW264.7 cells were pretreated with the compound (5, 10, and 20 μM) for 1h before exposure to LPS for 24 h. Protein samples for nuclear and cytosol extract of RAW264.7 cells were analyzed by Western blot using anti-Nrf2 antibody. The representative blots (**A**) and the calculated intensities (**B**) after normalization to Lamin B (nucleus) or β-actin (cytosol) were presented. Results are presented as the mean ± standard deviation from three independent experiments. The results differ significantly from LPS-only treated; * *p* < 0.05, ** *p* < 0.01, and *** *p* < 0.001.

**Table 1 molecules-28-01518-t001:** The ^1^H (500 MHz) and ^13^C NMR (125 MHz) data for **4** and **5** in CD_3_OD (δ in ppm, *J* values in parentheses).

no.	4			
	δ_C_	δ_H,_ Mult(*J* Hz)	δ_C_	δ_H,_ Mult(*J* Hz)
1	80.3, CH	3.74, dd(9.5, 6.6)	79.9, CH	3.67, dd(9.8, 6.9)
2	29.6, CH_2_	α; 2.29, br d(18.3) β; 1.44, ddt(18.3, 9.5, 3.4)	30.0, CH_2_	α; 2.26, m β; 1.34, m
3	123.1, CH	5.61, br s	131.6, CH	5.62, br s
4	137.4, C		130.4, C	
5	50.5, CH	2.45, dd(11.0, 2.2)	50.3, CH	2.27, m
6	82.9, CH	3.99, t(11.0)	82.0, CH	3.16, t(11.0)
7	52.0, CH	2.51, td(11.0, 3.2)	51.7, CH	2.29, m
8	22.1, CH_2_	α;2.02, br d(13.2) β; 1.56, qd(13.2, 3.4)	21.9, CH_2_	α;1.90, br d(13.5) β; 1.30, m
9	35.5, CH_2_	α;1.34, td(13.2, 3.4) β; 2.02, br d(13.2)	35.6, CH_2_	α;1.22, td(13.5, 3.4) β; 1.92, br d(13.5)
10	41.1, C		40.9, C	
11	140.8, C		140.6, C	
12	172.5, C		172.4, C	
13	117.3, CH_2_	5.47, d(3.2); 5.98, d(3.2)	117.3, CH_2_	5.42, d(3.4); 5.95, d(3.4)
14	12.3, CH_3_	0.67, s	12.4, CH_3_	0.45, s
15	65.3, CH_2_	4.07, d(15.2); 4.11, d(15.2)	68.9, CH_2_	4.39, d(11.5); 4.61, d(11.5)
1′	98.8, CH	4.44, d(7.8)	98.7, CH	4.40, d(7.8)
2′	75.6, CH	4.66, dd(9.8, 7.8)	75.5, CH	4.63, dd(9.5, 7.8)
3′	76.1, CH	3.54, dd(9.8, 8.1)	76.0, CH	3.53, dd(9.5, 9.5)
4′	71.9, CH	3.32, dd(9.8, 8.1)	71.9, CH	3.30, dd(9.8, 9.5)
5′	78.0, CH	3.26, ddd(9.8, 6.1, 2.0)	78.0, CH	3.24, ddd(9.8, 5.9, 2.0)
6′	62.8, CH_2_	3.65, dd(12.0, 6.1) 3.86, dd(12.0, 2.0)	62.8, CH_2_	3.63, dd(12.0, 5.9) 3.85, dd(12.0, 2.0)
1″			126.4, C	
2″,6″			131.7, CH	7.00, d(8.6)
3″,5″			116.5, CH	6.61, d(8.6)
4″			157.3, C	
7″			41.9, CH_2_	3.45, d(15.2); 3.53, d(15.2)
8″			173.2, C	
1‴	126.2, C		126.2, C	
2‴,6‴	131.6, CH	7.11, d(8.6)	131.6, CH	7.12, d(8.6)
3‴,5‴	116.3, CH	6.70, d(8.6)	116.3, CH	6.73 d(8.6)
4‴	157.6, C		157.8, C	
7‴	41.6, CH_2_	3.54, d(14.7); 3.58, d(14.7)	41.7, CH_2_	3.53, d(11.0); 3.56, d(11.0)
8‴	172.8, C		172.7, C	

## Data Availability

Not Applicable.

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
