# Peer review of "Sesquiterpene Lactones with Anti-Inflammatory Activity from the Halophyte Sonchus brachyotus DC"

_molecules, 2023, doi:10.3390/molecules28041518_

Round 1

Reviewer 1 Report

1. The fingerprint of S. brachyotus should be established and recognized the five sesquiterpene lactones in fingerprint.

2. In the bioactivity assay, a positive control should be used and comparison of 1.

Author Response

Reviewer 1

Thank you for your kind comments. We listed our response to your comments.

Comments and Suggestions for Authors

  1. The fingerprint of S. brachyotusshould be established and recognized the five sesquiterpene lactones in fingerprint.

→ Thank you for your comments. What do you mean by fingerprint is the chromatogram for the extract? If it’s correct, a fingerprint of the BuOH fraction containing compounds 1~5 was added in the supporting information by expressing the peak corresponding to the compound.

  1. In the bioactivity assay, a positive control should be used and comparison of 1.

→ As pointed out by reviewers, in the study to identify the action mechanisms of anti-inflammatory activity of materials, the positive controls are generally applied. Positive controls that is generally used, L-NAME (non-selective NOS inhibitor), AMT (iNOS inhibitor), Celecoxib (COX-2 inhibitor), dexamethasone (Interleukin receptor inhibitor) acts selectively on target proteins and/or cytokines participating in inflammatory response. This study aimed to identify the constituents of S. brachyotus and discover their new bioactivities. Here we reported the potent anti-inflammatory activities of new compounds isolated from S. brachyotus for the first time. Unfortunately, a positive control was not applied in this study mainly due to the target protein or cytokine on which the isolated compound acts was not specified. The inhibitory activity of the compounds was expressed as relative activity compared to the group treated with LPS only. However, we sympathize with the need for using a positive control in the assay, and in the further in-depth study identifying the detailed action mechanisms of the isolates, we will select an appropriate drug as positive control and compare the activities.

Reviewer 2 Report

Jeong, Rho, and co-workers have reported bioactivity guided isolation of two known and three unknown sesquiterpene lactones of the eudesmanolide class from the halophyte Sonchus brachyotus DC. They have determined the structure of the compounds using NMR spectroscopy and mass spectrometry. The compounds were identified as (1R, 5R, 6R, 7S, 10S)-5a,6bH-eudesma-3, 11(13)-dien-12,6a-olide and their different glycoside derivatives. Compounds 3 and 4 are isomeric with a p-hydroxyphenylacetyl group substitution at the aglycone and glycone hydroxyl groups, respectively. Compound 5 was substituted by two p-hydroxyhphenylacetyl groups, both glycone and aglycone parts. All the compounds are evaluated for their anti-inflammatory activity on LPS-activated RAW264.7 macrophages. Compound 1, without any glycoside attachment, suppressed the expression of iNOS and COS-2 as well as the production of TNF-a, IL-6, and IL-10 significantly and regulates the Nrf2/HO-1 pathway. Based on the above-mentioned results, they conclude that the native halophyte S. brachyotus containing bioactive sesquiterpenes, can be useful as a therapeutic agent for the treatment of inflammation-related diseases.

Comments: Considering the isolation and structure determination of several new compounds and the detection of anti-inflammatory activity in one of the compounds, the work reported in the manuscript has sufficient novelty for a new publication. However, the following concerns must be adequately addressed before accepting the manuscript for publication in Molecules:

i) The spectral interpretation for the structure determination of new compounds is not adequately described or properly presented in the manuscript.

a. I am sure authors have recorded both the DEPT 90 and DEPT 135 spectra of the compounds. DEPT spectra for all the new compounds should be presented in Supporting Information.

b. Presentation of NOESY spectrum data in the manuscript with corresponding assignment of through-space interaction on compound structure in a figure will be helpful for reader. Similarly, the presentation of NOESY spectrum in Supporting Information with compound structure and corresponding interaction in the inset can be helpful.

c. 13C NMR spectra of compounds 1 and 2 in pages 4 & 5 of the supporting information is not at all clear. For example, in 13C NMR of compound 2, the authors have considered a small peak at 172 as a quaternary carbon peak for ester, but ignored similar/higher intensity peaks at 127 and many in aliphatic region. A better 13C NMR spectrum of compound 2 with a higher signal-to-noise ratio must be presented.

d. The HSQC spectrum of compound 2 shows several unrelated extra peaks. That is confusing and raises questions about the credibility of determined structures. Authors must look into it and make necessary modifications before submitting a revised manuscript.

ii) There are several typographical errors in the manuscript, and those should be corrected before submission. e.g., in line 253.

Author Response

Thank you for your valuable comments. As you pointed out, the NMR spectra for compounds 1 and 2 were not clean. The compounds may be decomposed during storage in the MeOH solvent. This time, the two compounds were repurified and their NMR spectra were measured again.    

  1. i) The spectral interpretation for the structure determination of new compounds is not adequately described or properly presented in the manuscript.
  2. I am sure authors have recorded both the DEPT 90 and DEPT 135 spectra of the compounds. DEPT spectra for all the new compounds should be presented in Supporting Information.

→ Information from the DEPT spectrum can be found in the edited HSQC spectrum. All HSQC spectra in this study were obtained with the edited HSQC manner as provided in the supporting information. Therefore, the DEPT information for all new compounds could be unambiguously interpreted from the HSQC spectra.

  1. Presentation of NOESY spectrum data in the manuscript with corresponding assignment of through-space interaction on compound structure in a figure will be helpful for reader. Similarly, the presentation of NOESY spectrum in Supporting Information with compound structure and corresponding interaction in the inset can be helpful.

→ We added Figure 2 which shows the ROE correlations in the manuscript and indicated the correlations in the ROESY spectrum in the Supporting information.

  1. 13C NMR spectra of compounds 1 and 2 in pages 4 & 5 of the supporting information is not at all clear. For example, in 13C NMR of compound 2, the authors have considered a small peak at 172 as a quaternary carbon peak for ester, but ignored similar/higher intensity peaks at 127 and many in aliphatic region. A better 13C NMR spectrum of compound 2 with a higher signal-to-noise ratio must be presented.

→ Good suggestion. The two compounds appear to have slightly decomposed. We purified the two compounds again and measured the corresponding spectra. Improved spectra were presented in the supporting information.

  1. The HSQC spectrum of compound 2 shows several unrelated extra peaks. That is confusing and raises questions about the credibility of determined structures. Authors must look into it and make necessary modifications before submitting a revised manuscript.

→ As indicated above, the 2D spectra of the purified compound 2 were measured and presented in the supporting information.

  1. ii) There are several typographical errors in the manuscript, and those should be corrected before submission. e.g., in line 253.

→ We did our best to reduce typeographical errors. We did not really find any mistyped errors. We requested to edit and correct the English of the manuscript to a professional editing institute. There doesn’t seem to be any misspelled words in line 253.

Reviewer 3 Report

In this manuscript titled "Sesquiterpene lactones with anti-inflammatory activity from the Halophyte Sonchus brachyotus DC " - the methods of  1D and 2D NMR spectra, MS data, and optical rotation values of some lactones is used. The contents of the reviewed manuscript are very well described by the title. The authors show that compound 1 has the most potent activity.
In my point, the main message of this manuscript is very nice sound and could be recommended for publication in Molecules after a corresponding revision in which the following issues are addressed:
"Based on the NOESY spectrum and coupling constants, the configura" Page 2 Line 52 In your study, you are talking only about ROESY. What do you mean NOESY coupling constants ?
The authors have to shed light on the similarities and differences among their work and the literature of the problems anti-inflammatory activity. It has recently been shown that the action of anti-inflammatory  drug compounds may be due to a non-cyclogenase mechanism via the lipid membrane [https://doi.org/10.1016/j.molliq.2022.120502]. It is necessary to add a comperison of your results with the literature to the text of the manuscript.
A methods section should be extended to include more details obout NMR and 2D NMR experiments - the parameters of pulse sequence (width of spectrum, values of delay, standart of chemical shift, mixing times, etc ) used must be discription.
If the authors submit a new version of manuscript according to my suggestions with more explanations about the experimental details and additional discussions, I could recommend the manuscript for publication.

Author Response

Thank you for your kind comments. Based on your comments, we have revised or supplemented the manuscript.

1) "Based on the NOESY spectrum and coupling constants, the configura" Page 2 Line 52 In your study, you are talking only about ROESY. What do you mean NOESY coupling constants ?

→ You seem to have misunderstood. Not ROESY coupling constants, but proton coupling constants

To be apparent, the expression of “coupling constants” was replaced with “proton coupling constants in the 1H NMR spectrum”.

2) The authors have to shed light on the similarities and differences among their work and the literature of the problems anti-inflammatory activity. It has recently been shown that the action of anti-inflammatory drug compounds may be due to a non-cyclogenase mechanism via the lipid membrane [https://doi.org/10.1016/j.molliq.2022.120502]. It is necessary to add a comparison of your results with the literature to the text of the manuscript.

→ Thank you for your comments. Though we tested the inhibition of COX-2, we did not study the interaction with lipid membranes. Instead, in accordance to the Reviewer's comments, the recent report on the interaction of COX inhibitors with lipid membrane is additionally described in the text. (line 150-156, in page 5).

3) A methods section should be extended to include more details about NMR and 2D NMR experiments - the parameters of pulse sequence (width of spectrum, values of delay, standard of chemical shift, mixing times, etc ) used must be discription.

→ We added parameters about 2D NMR experiments in the method section.

Round 2

Reviewer 2 Report

The authors have made most of the necessary revisions in the main text and in the supporting information of the manuscript. The manuscript may now be accepted for publication in Molecules.